# A Multi-Technique Investigation of the Complex Formation Equilibria between Bis-Deferiprone Derivatives and Oxidovanadium (IV)

**DOI:** 10.3390/molecules27051555

**Published:** 2022-02-25

**Authors:** Rosita Cappai, Alessandra Fantasia, Guido Crisponi, Eugenio Garribba, M. Amélia Santos, Valeria Marina Nurchi

**Affiliations:** 1Dipartimento di Scienze della Vita e dell’Ambiente, Università di Cagliari, 09042 Monserrato-Cagliari, Italy; cappai@unica.it (R.C.); fantasia.91@hotmail.it (A.F.); crisponi@unica.it (G.C.); 2Dipartimento di Scienze Mediche, Chirurgiche e Sperimentali, Università di Sassari, Viale San Pietro, 07100 Sassari, Italy; garribba@uniss.it; 3Centro de Quimica Estrutural, Institute of Molecular Sciences, Instituto Superior Técnico, Universidade de Lisboa, Av. Rovisco Pais 1, 1049-001 Lisboa, Portugal; masantos@tecnico.ulisboa.pt

**Keywords:** oxidovanadium (IV), deferiprone, potentiometry, UV spectrophotometry, EPR spectroscopy

## Abstract

The increasing biomedical interest in high-stability oxidovanadium(IV) complexes with hydroxypyridinone ligands leads us to investigate the complex formation equilibria of V^IV^O^2+^ ion with a tetradentate ligand, named KC21, which contains two 3-hydroxy-1,2-dimethylpyridin-4(1*H*)-one (deferiprone) moieties, and with the simple bidentate ligand that constitutes the basic unit of KC21, for comparison, named L5. These equilibrium studies were conducted with joined potentiometric–spectrophotometric titrations, and the results were substantiated with EPR measurements at variable pH values. This multi-technique study gave evidence of the formation of an extremely stable 1:1 complex between KC21 and oxidovanadium(IV) at a physiological pH, which could find promising pharmacological applications.

## 1. Introduction

Vanadium compounds exhibit in humans different pharmacological activities, such as antiviral, antibacterial, anticancer and antidiabetic properties [1,2,3,4,5,6,7,8]. The historical accumulation of knowledge regarding vanadium antidiabetic activity is presented in an excellent review by Shechter et al. [9]. These authors report that, since its discovery (in 1980), vanadium salts could mimic the action of insulin on hexose uptake and glucose metabolism in rats, and intensive studies were carried out the insulinomimetic effects of new vanadium compounds and on the key mechanisms involved. More recently, research has been focused on the development of new coordinated ligands that could facilitate the absorption of the complexes by the gastro-intestinal tract and their transport in the blood. In this way, vanadium uptake by cells shows a potentially insulinomimetic and also anticancer activity [1,10,11]. At the moment, the reference molecules for any new vanadium compound characterized by antidiabetic activity are bis(maltolato)oxidovanadium(IV) (BMOV) and bis(ethylmaltolato)oxidovanadium(IV) (BEOV), with BEOV being the only vanadium compound submitted to clinical trials. Renal problems in some participating patients led to the withdrawal of these trials in phase II [12,13]. The antidiabetic properties of other V^IV^ compounds have been the focus of a large number of studies. Promising compounds in this regard are those formed by 3-hydroxy-1,2-dimethylpyridin-4(1*H*)-one (deferiprone, DFP) [14]. In particular, the complex [V^IV^O(DFP)_2_] is effective in the release of free fatty acids from isolated rat adipocytes, has better antidiabetic action than inorganic salt V^IV^OSO_4_ and presents a peculiar insulin-enhancing activity, as well as a noteworthy anticancer activity [15]. It shows a high antiproliferative activity against A375 malignant melanoma cells, causing apoptosis and cell cycle blocking. Moreover, it reduces the phosphorylation of ERK by about 80%, causing the deactivation of MAPK pathway in A375 cells, and induces the dephosphorylation of the retinoblastoma protein by 90% [16]. The complex formation equilibria of DFP with oxidovanadium(IV) (V^IV^O^2+^) ion were already characterized at 25 °C and 0.2 M KCl ionic strength by Buglyó et al. [17], who showed that the neutral complex VOL_2_ is the major species completely formed in the wide pH range of 5–9. Based on the above observations, we present here the characterization of the complex formation equilibria of the ligand 4-amino-7-((2-(3-hydroxy-2-methyl-4-oxopyridin-1(4*H*)-yl)propyl)amino)-4-(3-((2-(3-hydroxy-2-methyl-4-oxopyridin-1(4*H*)-yl)propyl)amino)-3-oxopropyl)-7-oxoheptanoic acid (KC21) (Figure 1) with V^IV^O^2+^. This ligand contains two 3-hydroxy-4-pyridinones (3,4-HP) units, attached to a backbone unity with the aim of remarkably increasing the thermodynamic and kinetic stability of the vanadium complex, because two chelating groups are needed to complete the V^IV^O^2+^ coordination sphere. Additionally, as compared with other reported bis(3,4-HP) ligands, the KC21 anchoring unit is an amino–carboxylic group with the capacity to interact with blood proteins, or even to be easily extra-functionalized to provide specific targeting capacity [18]. The synthesis of KC21, its acid base properties and its coordination ability toward Fe(III) and Zn(II) were recently reported [18]. Furthermore, for comparison purposes, we present herein the study of the complex formation equilibria of V^IV^O^2+^ ion with 1-(3′-aminopropyl)-3-hydroxy-2-methyl-4-pyridinone (L5), the mono-(3,4-HP) attaching arm of KC21 (Figure 1), whose synthesis and characterization were reported by Santos et al. [19].

## 2. Results

### 2.1. Protonation Equilibria

The protonation equilibria of the L5 ligand were previously studied via potentiometry at a ligand concentration of 2 × 10^−3^ M, 0.1 M KNO_3_ ionic strength and at 25 °C [19], while in the present work they were studied by combined potentiometric–spectrophotometric titrations. The treatment of experimental data with the HypSpec program [20] led to the protonation constants and to the corresponding speciation plots, reported in Table 1 and Figure 2, respectively. In the following, we briefly discuss the main features of absorptivity spectra (Figure 3A). The completely deprotonated species L^−^ is characterized by two bands, at 310 and 228 nm. The first protonation (log*K* 10.699), related to the NH_2_ group, leads to a slight decrease in these two bands and to the formation of a new band at 280 nm. These small spectral variations are reasonably connected to a micro speciation equilibrium between the NH_2_ and the O^–^ groups. The following protonation step (LH_2_^+^), taking place on the O^–^ group, leads to the disappearance of the band at 310 nm and to the rise of the band at 280 nm. In the last protonated species (LH_3_^2+^), the heterocyclic nitrogen atom is mainly protonated [17], the band at 280 nm strongly decreases with a slight blue shift and a small band forms at 245 nm. The log*K* values of the second and third protonation steps (9.494 and 3.142) are, respectively, 0.32 and 0.52 log*K* units lower than those calculated for the analogous log*K* values of DFP by Nurchi et al. [21] (9.494 vs. 9.82 and 3.142 vs. 3.66) (Table 1). These small differences can be associated with the positively charged species, which favor the proton release.

The protonation equilibria of KC21 (Table 1) were previously discussed in [18] based on potentiometry, UV, ^1^H and ^13^C NMR spectroscopy (log*K*_1_ = 10.43 and log*K*_2_ = 9.76 for O^−^ groups of the ring; log*K*_3_ = 8.68 for NH_2_ of the linker; log*K*_4_ = 4.48 and log*K*_5_ = 3.32 for the pyridinone nitrogen atoms; log*K*_6_ = 3.13 for the COO^–^ of the linker). The related absorptivity spectra are reported in Figure 3B.

### 2.2. Oxidovanadium(IV) Complex Formation Equilibria

The equilibria of complex formation of V^IV^O^2+^ with L5 and KC21 ligands were studied using combined potentiometric–spectrophotometric titrations supported by EPR measurements at 1:1 and 1:2 V^IV^O^2+^:ligand molar ratios. The complex formation constants are reported in Table 2 and the related speciation plots in Figure 4. The calculated absorptivity spectra of V^IV^O^2+^-L5 depicted in Figure 5A give evidence for the formation of the 1:1 species [(V^IV^O)LH]^2+^, where V^IV^O^2+^ is coordinated by one L5 unit, and of the major 1:2 species [(V^IV^O)L_2_H_2_]^2+^, in which a second L5 unit enters in the coordination. The speciation model (Figure 4) strictly resembles that reported for DFP ([(V^IV^O)L]^+^ with log*K* = 12.18, [(V^IV^)OL_2_] with log*K* = 22.83, [(V^IV^O)L_2_H_−1_]^−^ with log*K* = 12.43 and [(V^IV^O)_2_L_3_H_2_]^+^ with log*K* = 38.5) [17]. The species [(V^IV^O)L]^2+^ is now replaced by [(V^IV^O)LH]^2+^ and [(V^IV^O)L_2_] by [(V^IV^O)L_2_H_2_]^2+^. As mentioned in the discussion of the protonation equilibria, this behavior can be attributed to the presence of positively charged species that is also reflected in the pM differences (11.7 for L5 vs. 13.9 for DFP).

Concerning the V^IV^O^2+^-KC21 complexes, during the spectrophotometric titration at 1:1 V^IV^O^2+^:ligand molar ratio, some turbidity was observed above pH 3, giving dirty spectra. Therefore, only the data collected at 1:2 V^IV^O^2+^:ligand molar ratio were used in the HypSpec treatment. The UV–spectrophotometric data (Figure 5B) show the formation of differently protonated 1:1 complexes whose complex formation constants and speciation plot are reported in Table 2 and Figure 4. The formation of the [(V^IV^O)LH_4_]^3+^ complex starts at extremely acidic conditions (pH 0), reaching a maximum at pH 1.7; in this complex, V^IV^O^2+^ is presumably coordinated by two oxygen atoms of one 3,4-HP moiety, predominantly by the second until it is protonated on O^–^ and N atoms, as well as on the NH_2_ and COO^–^. This is substantiated by the decrease in the band at 280 nm related to the protonated O^–^ in the 3,4-HP moiety. After pH 2, the formation of [(V^IV^O)LH_2_]^+^ takes place, reaching its maximum at pH 2.5, where V^IV^O^2+^ is coordinated by both the 3,4-HP moieties, while the ligand is still protonated on the NH_2_ and COO^–^. The loss of a proton comes at about 0.3 units lower (pK 2.84) than that in the free ligand. A further proton is lost at pK 5.92, presumably from the charged NH_3_^+^, since it does not give significant spectral variation, as would be expected for the deprotonation of an amino group. The last deprotonation (pK 10.13) concerns the coordinated water to give [(V^IV^O)LH_−1_]^2−^.

### 2.3. EPR

The EPR spectra of the V^IV^O^2+^-L5 system collected at the 1:2 V^IV^O^2+^:ligand molar ratio at increasing pH (Figure 6A) show the formation of 1:2 complexes.

The [(V^IV^O)LH]^2+^ species is observed in the pH range of 2–3. Its spin Hamiltonian parameters are *g_z_* = 1.938 and *A_z_* = 171.0 × 10^−4^ cm^−1^ (the *M*_I_ = 7/2 resonance of this species is denoted with **I**). From pH 3, the formation of the major species [(V^IV^O)L_2_H_2_]^2+^ is observed up to pH 10.5, with the amine group still protonated. In the range of existence of [(V^IV^O)L_2_H_2_]^2+^, the EPR spectra show two sets of resonances (**IIa** and **IIb**), suggesting the presence of two species. The ratio between the intensity of the spectral signals, which remains constant along the pH range 5-10, indicates that the two complexes are in equilibrium. The spin Hamiltonian parameters are *g_z_* = 1.940 and *A_z_* = 166.6 × 10^−4^ cm^−1^ for **IIa**, and *g_z_* = 1.950 and *A_z_* = 159.4 × 10^−4^ cm^−1^ for **IIb**. Based on an “additivity relationship” [23,24], these values are compatible with a *cis*-octahedral V^IV^O^2+^ complex with an (equatorial–equatorial) and an (equatorial–axial) arrangement of the two L5 units with respect to the V=O bond, and a water molecule that completes the equatorial coordination sphere (**IIa**) and a square pyramidal species with 2 × (equatorial–equatorial) arrangements (**IIb**). The *A_z_* values, slightly lower than that expected from the “additivity relationship”, can be explained assuming that hydroxypyridinones have an electronic structure intermediate between the hydroxypyrones and catechols, with a partial positive charge on the N atom in position 1 of the ring and a pseudo-aromatic electronic structure that approaches that of 1,2-dihydroxybenzene [25]. This behavior is similar to DFP [17] and other pyridinones derivatives [26] and confirms that the terminal amino donor does not take part in the metal coordination. At pH > 10, the EPR intensity significantly decreases due to the hydrolytic processes and formation of the hydroxide species [(V^IV^O)_2_(OH)_5_]^–^, EPR-silent.

The EPR spectra of KC21 presented in Figure 6B suggest that the complexation starts around pH 1 with the formation of the species [(V^IV^O)LH_4_]^3+^ (whose *M*_I_ = 7/2 resonance is indicated with **I**), where only one (CO, O^–^) group binding the V^IV^O^2+^ ion, presumably [(V^IV^O)LH_4_]^3+^, with the amino and pyridine nitrogen, the second -OH and COOH still protonated. The coordination mode remains the same, (CO, O^–^), in the transient species [(V^IV^O)LH_2_]^+^. With the following two deprotonations, [(V^IV^O)LH] and [(V^IV^O)L]^–^, both the (CO, O^–^) groups can interact with V^IV^O^2+^ ion. In this case, also, two sets of resonances are detected (**IIa** and **IIb**), indicating the presence of two species in equilibrium, the *cis*-octahedral and the square pyramidal complexes, shown in Figure 7. The EPR parameters are similar to those measured for L5 (Table 3). Notably, the relative amount of **IIa** compared to **IIb** is larger than that with L5. This observation can be explained bythe high steric constraints associated with the binding of the two (CO, O^–^) groups of KC21 on the equatorial plane; instead, in the *cis*-octahedral complex the arrangement (eq-eq; eq-ax) results in a more relaxed structure. After pH 9, the species **III** is detected. Its spin Hamiltonian parameters are *g_z_* = 1.943 and *A_z_* = 162.2 × 10^−4^ cm^−1^ and can be ascribed to a hydroxide complex upon the deprotonation of the water ligand, [(V^IV^O)LH_−1_]^2−^, in agreement with thermodynamic results. In such a complex, the two (CO, O^–^) functions should be arranged in an (eq-eq) and (eq-ax) mode, and an OH^–^ ion should occupy the fourth equatorial site. As observed in the literature [17,26], the formation of this complex shifts to the right the equilibrium square pyramidal ⇄ *cis*-octahedral species. The pK for the deprotonation of H_2_O ligand, in the pH range 9–10, is comparable to those reported for other pyridinones. Moreover, the decrease in *A_z_* (~4 × 10^−4^ cm^−1^) is compatible with the transformation of a water into a hydroxide ligand.

## 3. Materials and Methods

### 3.1. Reagents

NaOH, NaCl, HCl and VOSO_4_·3H_2_O were Sigma Aldrich products, and all the reagents were used without any further purification. The L5 and KC21 ligands were synthetized as previously reported [18,19]. Oxidovanadium(IV) sulfate solution ~0.1 M was prepared weekly, acidified with a stoichiometric amount of HCl to prevent hydrolysis and standardized by redox titration [27,28]. All solutions were prepared using grade A glassware and ultrapure water (conductivity < 0.1 μS).

### 3.2. Solution Equilibria Studies

The complex formation equilibria were studied at 25 °C and 0.1 M NaCl ionic strength through combined potentiometric–spectrophotometric titrations at 1:1 and 1:2 V^IV^O^2+^:ligand molar ratios and a constant ligand concentration 3.0 × 10^−4^ M for L5 and 5.0 × 10^−4^ M for KC21. Potentiometric measurements were performed with a dEcotrode plus Metrohm combined glass electrode connected to 888 Titrando (Metrohm AG, Herisau, Switzerland). The electrode was calibrated daily for hydrogen ion concentration via HCl standard titration with NaOH in the used experimental conditions, and data were analyzed using Gran’s method [29]. Spectrophotometric measurements were performed in the 200–400 nm wavelength range with a 0.2 cm fiber optic dip probe connected to an Agilent Cary 60 UV-vis spectrophotometer. Potentiometric and spectrophotometric data were processed using the HyperQuad and HypSpec programs, respectively [20,30]. Log β_pqr_ values refer to the overall equilibria *p*M + *q*H + *r*L ⇆ M_p_H_q_L_r_ (electrical charges omitted). The following hydrolysis constants of V^IV^O^2+^ were assumed for the calculations: [V^IV^O(OH)]^+^ (logβ_1–1_ = −5.94), [(V^IV^O)_2_(OH)_2_]^2+^ (logβ_2–2_ = −6.95), [31] [V^IV^O(OH)_3_]− (logβ_1–3_ = −18.0) and [(V^IV^O)_2_(OH)_5_]^−^ (logβ _2–5_ = −22.0) [32,33].

### 3.3. EPR Experiments

The solutions were prepared by dissolving in ultrapure water (from a millipore Milli-Q Academic purification system) a weighed amount of VOSO_4_∙3H_2_O and L5 or KC21 for a metal ion concentration of 2 mM and 1:1 and 1:2 V^IV^O^2+^:ligand molar ratios. The solutions were bubbled with argon in order to avoid the oxidation of the metal ion. The pH values were varied with diluted solution of H_2_SO_4_ and NaOH. DMSO was added to each sample to uniformly freeze the solutions and prevent a concentration gradient during freezing. Anisotropic EPR spectra were recorded at 120 K with an X-band Varian E-9 spectrometer equipped with a variable temperature unit. The microwave frequency was 9.15–9.16 GHz, microwave power 20 mW, time constant 0.5 s, modulation frequency 100 kHz and modulation amplitude 0.4 mT.

## 4. Conclusions

The use of the tetradentate ligand KC21, in which two deferiprone units are connected through a long and flexible linker, has led to the formation of an extremely stable 1:1 complex with a V^IV^O^2+^ ion. This formation of the neutral form (V^IV^O)LH that reaches its maximum near pH 4.5, the negatively charged form [(V^IV^O)L]^–^ that is stable in the pH range 6–10, and then of the hydroxo complex [(V^IV^O)LH_−1_]^2–^ is observed. The EPR spectra related to these complexes shows two sets of resonances indicative of the presence of two species in equilibrium, the *cis*-octahedral and the square pyramidal complexes. Remembering that pM gives the real evaluation of the ligand strength in its interaction with a given metal ion, and the extremely high pV^IV^O^2+^ value for a KC21 ligand 21.8 has to be pointed out. This value is about 8 units greater than the corresponding value of 13.9 found for the parent molecule deferiprone. A similar increase in the pM passing from DFP to KC21 was found for Fe^III^, Al^III^ and Zn^II^ (5.1, 3.2 and 2.1, respectively) [18]. This increase in stability can be accounted for an entropic stabilization of the complexes formed by a ligand bearing two different DFP units. In simplistic terms, the value of this increment is indicative of a favorable conformation of the ligand in the coordination of a given metal ion, and in this respect, the extraordinary efficiency of KC21 in V^IV^O^2+^ coordination has to be remarked. The good stability of [(V^IV^O)L]^–^ could positively influence its absorption and transport, minimizing the hydrolytic processes and precipitation of the hydroxide V^IV^O(OH)_2_ at the low vanadium concentration present in the biological fluids of humans treated with vanadium compounds (1–10 μM) [34,35,36,37,38]. The presence of the COOH and NH_2_ groups on the KC21 skeleton constitutes an essential feature of this ligand, since these groups can interact with blood proteins and facilitate the complex blood transport or cell uptake. On the other hand, its possible extra-functionalization can provide specific capacity for further biotargeting or grafting of this ligand/complex on solid nanomaterials, expanding the potential clinical applications of this chelating agent and its complexes. Overall, the present results show a new chemical platform for potent vanadium complex and encourage the progression of these studies with biochemical and biological assays. We envisage prospective medical applications as insulinomimetics or even as anticancer agents.

## Figures and Tables

**Figure 1 molecules-27-01555-f001:**
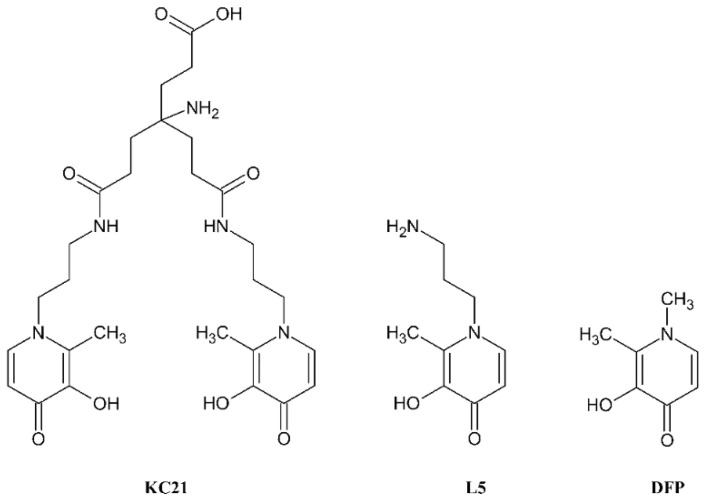
Chemical structures of the ligands KC21, L5 and DFP.

**Figure 2 molecules-27-01555-f002:**
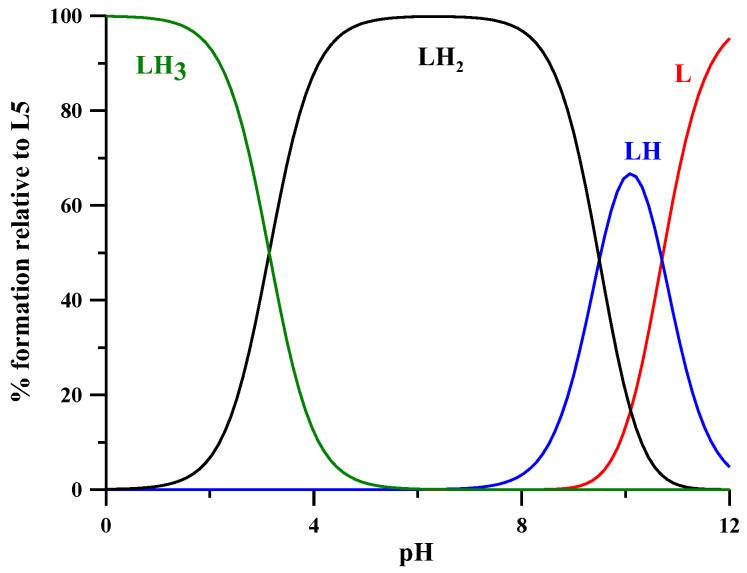
Speciation plot of L5 ligand calculated with HySS [22] according to the protonation constants in Table 1.

**Figure 3 molecules-27-01555-f003:**
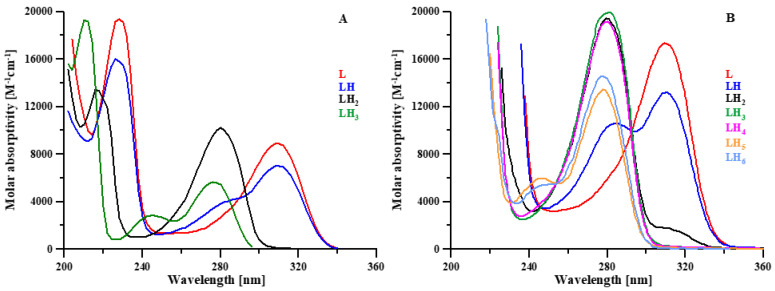
Calculated molar absorptivity spectra of the differently protonated forms of L5 (**A**) and KC21 (**B**) ligands.

**Figure 4 molecules-27-01555-f004:**
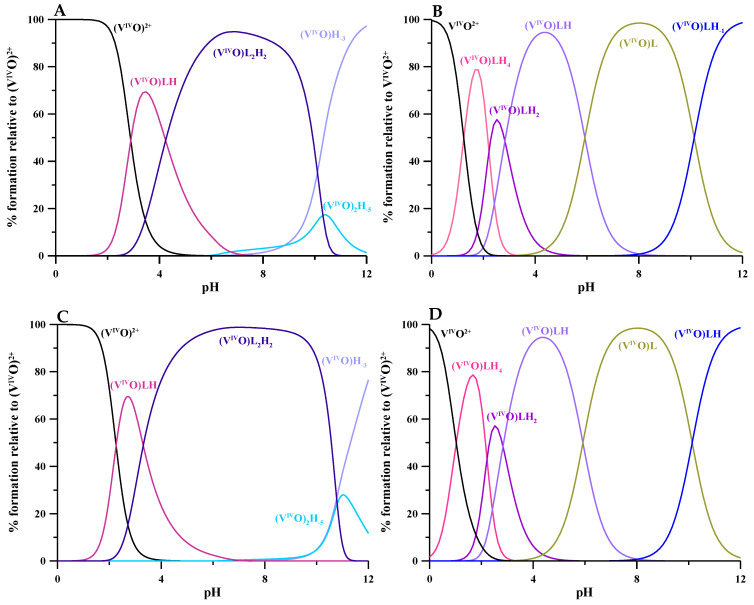
Speciation plots of V^IV^O^2+^-ligand systems calculated with HySS program [22]. Top: UV–spectrophotometric titration conditions, 1:2 V^IV^O^2+^:ligand molar ratio and ligand concentration 3 × 10^−4^ M and 5 × 10^−4^ M, respectively, for L5 and KC21, (**A**) L5, (**B**) KC21. Bottom: EPR measurements conditions, 1:2 and 1:1 V^IV^O^2+^:ligand molar ratio, respectively, for L5 and KC21 at V^IV^O^2+^ concentration 2 × 10^−3^ M, (**C**) L5, (**D**) KC21. Charges are omitted for simplicity.

**Figure 5 molecules-27-01555-f005:**
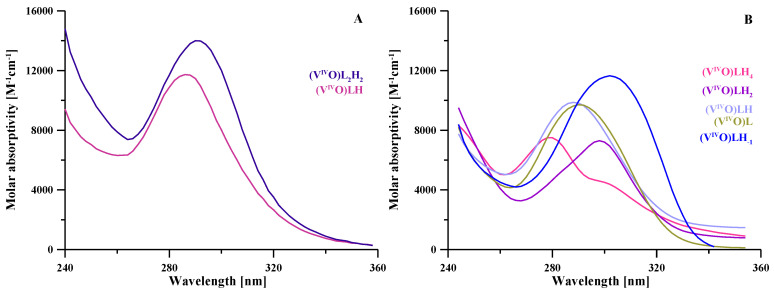
Calculated molar absorptivity spectra of differently protonated forms of V^IV^O^2+^ complexes with L5 (**A**) and KC21 (**B**) ligands.

**Figure 6 molecules-27-01555-f006:**
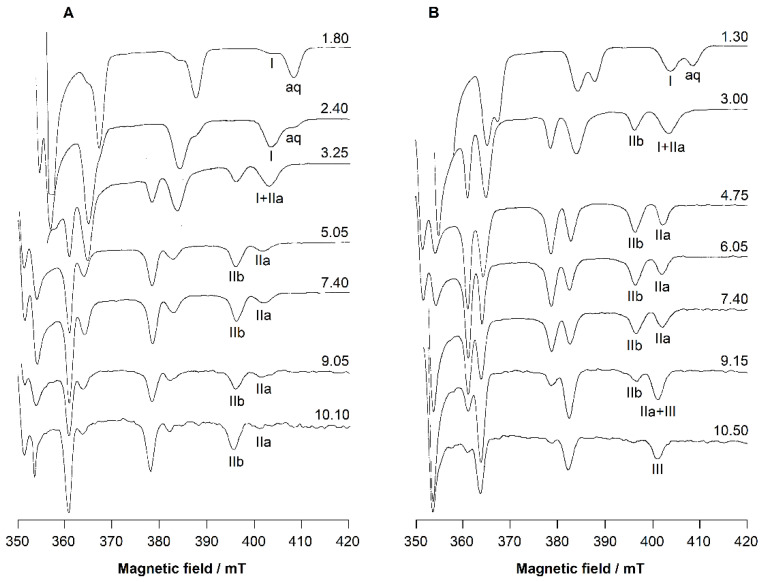
High field region of the anisotropic X-band EPR spectra, at increasing pH, of V^IV^O^2+^-L5 system at 1:2 V^IV^O^2+^:ligand molar ratio (**A**) and V^IV^O^2+^-KC21 system at 1:1 V^IV^O^2+^:ligand molar ratio (**B**). V^IV^O^2+^ concentration is 2 × 10^−3^ M.

**Figure 7 molecules-27-01555-f007:**
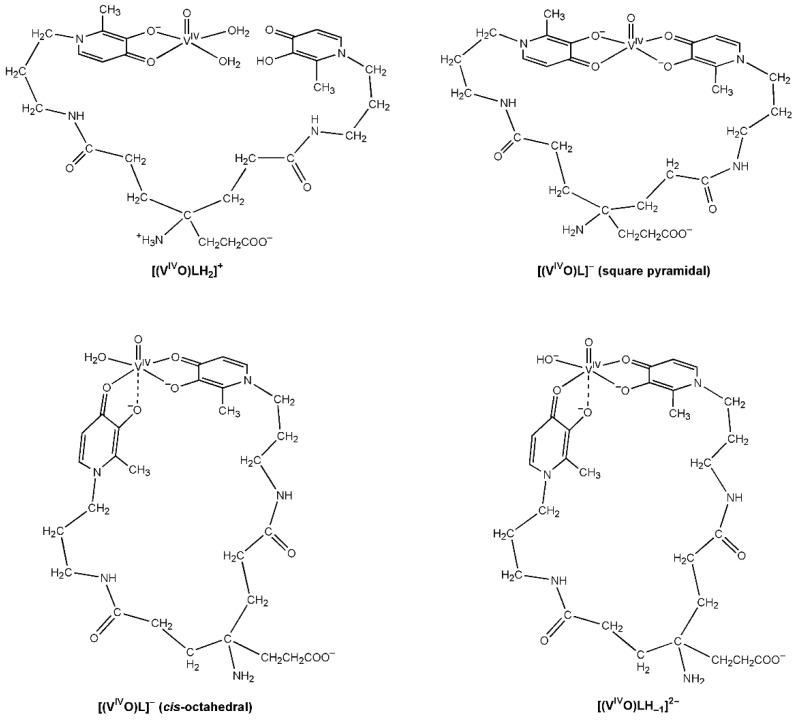
Molecular drawings of the V^IV^O^2+^ species with the ligand KC21 based on the solution studies.

**Table 1 molecules-27-01555-t001:** Protonation constants of KC21, L5 and DFP ligands evaluated from combined potentiometric–UV spectrophotometric titrations at 25 °C, 0.1 M ionic strength. L indicates the completely deprotonated form of ligands. The digits in brackets represent the standard deviation calculated by HypSpec program [20].

	KC21 [18]			L5			DFP [21]	
Species	logβ	log*K*	Species	logβ	log*K*	Species	logβ	log*K*
LH^2−^	10.43	10.43	LH	10.699 (2)	10.699	LH	9.82	9.82
LH_2_^−^	20.19	9.76	LH_2_^+^	20.193 (3)	9.494	LH_2_^+^	13.48	3.66
LH_3_	28.87	8.68	LH_3_^2+^	23.335 (7)	3.142			
LH_4_^+^	33.35	4.48						
LH_5_^2+^	36.67	3.32						
LH_6_^3+^	39.80	3.13						
Ionic strength	NaCl			NaCl			KCl	

**Table 2 molecules-27-01555-t002:** Complex formation constants and pM values of V^IV^O^2+^ with KC21, L5 and DFP ligands evaluated from combined potentiometric–UV titrations at 25 °C, 0.1 M NaCl ionic strength using HypSpec program [20]. pM = −log_10_[M_free_] with [M] = 10^−6^ M, [ligand] = 10^−5^M and pH = 7.4.

KC21		L5		DFP [17]	
Species	logβ	Species	Logβ	Species	Logβ
[(V^IV^O)LH_4_]^3+^	40.74(1)	[(V^IV^O)LH]^2+^	21.398(4)	[(V^IV^O)L]^+^	12.18
[(V^IV^O)LH_2_]^+^	36.31(1)	[(V^IV^O)L_2_H_2_]^2+^	41.48(1)	[(V^IV^O)L_2_]	22.83
[(V^IV^O)LH]	33.47(1)			[(V^IV^O)L_2_H_−1_]^−^	12.24
[(V^IV^O)L]^−^	27.55(1)			[(V^IV^O)L_3_H_2_]^+^	38.5
[(V^IV^O)LH_−1_]^2−^	17.42(4)			[(V^IV^O)_2_L_2_H_−2_]^2−^	16.43
p(V^IV^O^2+^)	21.8		11.7		13.91

**Table 3 molecules-27-01555-t003:** EPR spin Hamiltonian parameters for V^IV^O^2+^ complexes formed in aqueous solution by L5 and KC21.

Ligand	Complex	Symbol in Figure 6	*g_z_*	*A_z_ ^a^*	Arrangement of the Ligand Donors
L5	(V^IV^O)LH	**I**	1.938	171.0	eq-eq
	*cis*-(V^IV^O)L_2_H_2_	**IIa**	1.940	166.6	eq-eq; eq-ax
	(V^IV^O)L_2_H_2_	**IIb**	1.950	159.4	eq-eq; eq-eq
KC21	(V^IV^O)LH_4/_(V^IV^O)LH_2_	**I**	1.938	170.8	eq-eq
	*cis*-(V^IV^O)LH/(V^IV^O)L	**IIa**	1.940	166.8	eq-eq; eq-ax
	(V^IV^O)LH/(V^IV^O)L	**IIb**	1.951	159.1	eq-eq; eq-eq
	(V^IV^O)LH_−1_	**III**	1.943	162.2	eq-eq; eq-ax

^a^ *A_z_* in 10^–4^ cm^–1^.

## Data Availability

The data are available from the authors.

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
