# Peer review of "A Multi-Technique Investigation of the Complex Formation Equilibria between Bis-Deferiprone Derivatives and Oxidovanadium (IV)"

_molecules, 2022, doi:10.3390/molecules27051555_

Round 1

Reviewer 1 Report

The manuscript titled „A multi-technique investigation of the complex formation equilibria between bis-deferiprone derivatives and oxidovanadium(IV)” discusses solution equilibrium results. These results were collected by correct measurements and evaluations, but the manuscript is to be improved before it is recommended to publish

Comments to the manuscript:

  • Table 1 should contain all the protonation constants known for the three ligands (only mentioning them in the text is not a really good solution).  
  • Oxidovanadium(IV) complexes of two DFP derivatives, KC21 and L5, have been studied in the present work. Although, the results with the parent molecule, DFP, have been previously published by other authors (ref. 17), for a more clear comparison, those results (equilibrium model and values) should be involved into the Tables 1 and 2. A detailed evaluation of the 
  • In Ref. 17 also tris-complex, dinuclear species were found with DFP. Here these species are not mentioned. The reason of these differences should be evaluated.
  • What is the reason, why only 1:1 and 1:2 ratios were studied? 
  • Specifying of the values in Tables 1 and 2 with three decimal is surely not correct. Determination of that values with such accuracy is not realistic. Correction is suggested.
  • The conditions, used during the calculation of p(VIVO2+) should be given to Table2. Due to the high affinity of this metal cation for hydrolysis, the hydroxide ions might compete in a significant extent with the ligands for the oxidovanadium(IV) under the condition, where the p(VIVO2+) is calculated. This need to be taken into account. 

Author Response

  • The protonation constants for all the ligands are reported in Table 1.
  • The formation constants of VIVO2+ complexes for all the ligands are reported in Table 2.

3/4       Preliminary proofs at 1:3 and 1:4 molar ratios did not give evidence of complexes with stoichiometry higher than 1:2. The non-oxido tris-complex (indicated as VOL3H2, i.e. VL3xH2O, in ref. 17) and di-m-hydroxido dinuclear ((VO)2L2H–2 in ref. 17) are two minor species. When the distribution curves of the system VO2+-DFP at molar ratio 1:2 and vanadium concentration 1 mM are built, VOL3H2 reaches 3.0% and (VO)2L2H–2 less than 1%. As pointed out by Rangel et al. in ref. 26, to increase the amount of the tris-complex is necessary to isolate the solid bis-chelated species and use an excess of the ligand in CH3COOH that favors the loss of the oxido ligand of VO2+ ion as a water molecule.

In the system with KC21, the percent amount of these two species is still lower due to the steric hindrance of the ligand. In fact, to form the non-oxido complex, for example, two ligand molecules, one with both the donor set (CO, O–) and another with one (CO, O–) set have to simultaneously bind a 'bare' V(IV) ion with an unfavorable interaction. So, they are neglected in this study. In any case, their presence does not improve the fitting of the titration curves.

5          The protonation constants of L5 were evaluated from potentiometric-spectrophotometric results of high precision and accuracy, which allowed obtaining three significant digits.

6          The condition used in the calculation of pVIVO2+ were given in Table 2.

The pM (-log of free metal ion concentration calculated at pH 7.4, metal concentration 1x10-6 M and ligand concentration 1x10-5 M) is a conventional measure of the chelating strength of a ligand. The hydrolysis constant should be disregarded in its calculation. In fact, if taken into account, the values for the ligands with a pM less than that of hydrolysis alone become equal to that of hydrolysis.

Reviewer 2 Report

Nurchi and co-workers report on the complex formation equilibria between bis-deferiprone derivatives and oxidovanadium(IV) using various physicochemical techniques.  The manuscript constitutes an interesting contribution to the bioinorganic chemistry of vanadium and thus, it is my pleasure to recommend it for publication in Molecules, providing that the authors will address the following points.

Page 1,  line 30.  Please change discover  to discovery.

Page 2, line 64.  Please change Fe3+ and Zn2+ to FeIII and ZnII

Page 8, line 260.  Please change Fe3+ and Al3+ to FeIII and AlIII

Please provide molecular drawings of the oxidovanadium(IV) species with the ligand KC21on the basis of the solution studies.

Author Response

Thank you for your kind comments, and for the suggestions. We performed the requested revisions.

Reviewer 3 Report

The manuscript is well prepared, the experiments are clearly designed and well described. Moreover, the spectroscopic data is quite convincing and well presented. In a previous publication (ref 18), the authors reported on the complexation behaviour of KC21 towards Fe(III), Al(III) and Zn(II) ions. I think a paragraph with a short comparison between the complexation behaviour toward VO(2+) (this work) and previous findings would help putting the current results in relation. 

A minor typo I spotted is in line 30, introduction: discovery instead of discover.

Other than that I support publication in Molecules after very minor revisions

Author Response

Thank you for your kind comments. As suggested, in the Conclusions section we added a comparison between the results of complex formation of KC21 with VIVO2+ and the previous results obtained with FeIII, AlIII and ZnII.

Author Response

Thank you for your comments and for your suggestions that helped us to improve the quality of the paper. We checked accurately the manuscript for typographic errors, and revised the English language with the help of a native English speaker. In particular, we revised all the mistakes pointed out by you.

Round 2

Reviewer 1 Report

Accepting of the manuscript for publication is suggested. 

Author Response

Thank you for your kind comments.